# Datterino Trial: A Double Blind, Randomized, Controlled, Crossover, Clinical Trial on the Use of Hydroponic Cultivated Tomato Sauce in Systemic Nickel Allergy Syndrome

**DOI:** 10.3390/jcm11185459

**Published:** 2022-09-16

**Authors:** Angela Rizzi, Raffaella Chini, Serena Porcari, Carlo Romano Settanni, Eleonora Persichetti, Vincenzina Mora, Caterina Fanali, Alessia Leonetti, Giuseppe Parrinello, Franziska Michaela Lohmeyer, Riccardo Inchingolo, Maria Cristina Mele, Antonio Gasbarrini, Eleonora Nucera

**Affiliations:** 1UOSD Allergologia e Immunologia Clinica, Dipartimento Scienze Mediche e Chirurgiche, Fondazione Policlinico Universitario A. Gemelli IRCCS, 00168 Roma, Italy; 2UOC Medicina Interna e Gastroenterologia, CEMAD Centro Malattie dell’Apparato Digerente, Fondazione Policlinico Universitario Gemelli IRCCS, 00168 Roma, Italy; 3UOC Nutrizione Clinica, Dipartimento Scienze Mediche e Chirurgiche, Fondazione Policlinico Universitario A. Gemelli IRCCS, 00168 Roma, Italy; 4Direzione Scientifica, Fondazione Policlinico Universitario A. Gemelli IRCCS, 00168 Roma, Italy; 5UOC Pneumologia, Dipartimento Scienze Mediche e Chirurgiche, Fondazione Policlinico Universitario A. Gemelli IRCCS, 00168 Roma, Italy; 6Medicina e Chirurgia Traslazionale, Università Cattolica del Sacro Cuore, 00168 Roma, Italy; 7UOC Medicina Interna e Gastroenterologia, Dipartimento di Scienze Mediche e Chirurgiche, Fondazione Policlinico Universitario Gemelli IRCCS, 00168 Roma, Italy

**Keywords:** food allergy, Systemic Nickel Allergy Syndrome, low-nickel diet, tomato, hydroponic agriculture, symptoms, quality of life, adherence, *precision agriculture*, *precision medicine*

## Abstract

A low-nickel (Ni) diet, a key treatment for Systemic Nickel Allergy Syndrome (SNAS), is difficult in the long term and strongly impacts quality of life (QoL). Hydroponic agriculture could be an alternative to allow the reintroduction of tomato, an essential food in the global diet. In a first interventional, randomized, double-blind, single-center crossover study, we compared the possible effects of eating tomato puree deriving from hydroponic agriculture versus tomato puree from conventional cultivation, collecting data on subjective control of SNAS symptoms, adherence to treatment, and impact on QoL. Thirty subjects were randomly assigned to one of the following treatment groups: (1) a 12-week low-Ni diet plus 100% Italian Datterino tomato puree deriving from hydroponic technology; (2) a 12-week low-Ni diet plus 100% Italian Datterino tomato puree deriving from conventional cultivation. Then, after a 2-week washout period on the low-Ni diet, each patient crossed over to the other treatment. Patients reported lower symptom scores after eating Datterino tomato puree deriving from hydroponic technology; specifically, bloating (*p* = 0.0111, *p* = 0.0060), flatulence (*p* = 0.0090), abdominal cramps (*p* = 0.0207), constipation (*p* = 0.0395), and diarrhea (*p* = 0.0105). Overall, the adherence rate was high for both treatment arms. At baseline, QoL was poor, but significant improvement was observed after two treatments. In our study, *precision medicine* and *precision agriculture* merge in a holistic approach to the challenges of food allergies.

## 1. Introduction

Hydroponics, a term derived from the Greek words *hydro* (water) and *ponos* (labor), is a form of soilless culture. It refers to a technique in which the roots of plants are suspended in a static continuously aerated nutrient solution or in a continuous stream or mist of nutrient solution [1]. Its origins are ancient: The hanging gardens of Babylon and the floating gardens of the Aztecs in Mexico were hydroponic in nature. In the 1800s, the basic concepts for hydroponic plant cultivation were established by those who studied how plants grow [2]. In the 1930s, a Californian scientist popularized soilless plant culture through manuscripts still considered pillars of the science of hydroponics [3].

The hydroponic technique represents the greatest innovation of the recent past in the protected crops sector, which constitutes a solution for the cultivation of horticultural species when the characteristics of the soil make it impractical [4]. Furthermore, soilless cultivation drastically reduces the environmental impact owing to the efficient use of water and fertilizers and the limited release of pollutants into the environment. Interestingly, in the absence of soil, disease transmission and infections by insects or parasites are impossible, which reduces or eliminates the use of pesticides and their consequent toxicity. Finally, plants require less time to grow compared with the crop grown in the field [5].

Tomatoes (*Solanum lycopersicum*) are a basic product in many countries of the world, representing one of the most important vegetable crops in terms of production quantity and source of chemical compounds such as nutrients and anti-oxidants [6,7]. These vegetables are routinely grown in both soil and hydroponic systems. The growing literature supports the contribution of hydroponic techniques to improve food quality [8,9,10,11,12].

The tomato, in particular its concentrated form, is one of the major food vegetable sources of nickel (Ni). It is known that the average concentration of Ni in fresh tomatoes is 500 µg/Kg and higher than 500 µg/Kg in concentrated tomato [13,14].

Nickel is a silver-white metallic element widely diffused in nature. It is present in soil, water, and air and is widely used for industrial purposes [15]. The origin of its name derived from the German term *kupfernickel* (kupfer: copper; nickel: from Nicolaus evil genius or goblin, which according to the miners did not let them find copper in minerals). This metal is contained in many accessories and tools for daily use [15] and it is the major cause of Allergic Contact Dermatitis (ACD) [16].

The ingestion of Ni-rich foods causes systemic skin disorders (urticaria/angioedema, erythema, eczema) and extracutaneous multi-organ symptoms (heartburn, meteorism, abdominal pain, alvo, oral aphthosis, headache, respiratory disorders, recurrent infections, fibromyalgia), expressed singularly or in variable association in some particularly sensitive subjects [17]. This clinical picture is known as Systemic Nickel Allergy Syndrome (SNAS) [17], which occurs in approximately 20% of ACD patients [13,18].

The human body requires about 25–35 µg of Ni per day, but a common diet provides 200 to 600 µg of Ni per day (of which only 1–10% is absorbed) [19].

A low-Ni diet, following positive patch tests, in addition to being an effective diagnostic tool [13,20], represents an important first-line non-pharmacological therapeutic option in the control of systemic manifestations, resulting in significant clinical improvement [20,21,22,23,24].

However, such a nutritional approach can be difficult for several reasons such as: the variability of Ni concentration in the soil that varies according to soil type, synthetic fertilizer and pesticide use, and contamination of the soil by industrial effluents and municipal waste [25,26]. Equally variable is the Ni content in different batches of the same food depending on the season, the part of the plant (Ni tends to concentrate mainly in the leaves), and the age of the leaves in leafy foods that are consumed (greater Ni accumulation in older leaves) [25].

Controlled environment agriculture (CEA) can be a potential and sustainable alternative to conventional farming systems. The use of “closed-cycle” hydroponic methods in a completely controlled, aseptic, artificial, soil-free environment allows controlled production, from both a qualitative and a hygienic–sanitary point of view, which solves the problem of soil contamination, allowing obtaining vegetables free from harmful substances [8,27].

The potential of hydroponic techniques, the nutritional properties of tomato—a key vegetable in the Mediterranean diet—and the repercussions of its exclusion in the low-Ni diet constitute the rationale of our research project.

Consequently, we raised the question whether the intake of tomato puree deriving from hydroponic agriculture will help patients suffering from SNAS to control their symptoms and to improve their quality of life (QoL).

## 2. Materials and Methods

### 2.1. Design and Trial Objectives

We conducted the first interventional, randomized, double-blind, single-center crossover controlled clinical trial on the clinical impact of hydroponic cultivated tomato sauce in a cohort of SNAS patients following a low-Ni diet for at least 4–6 weeks.

The primary outcome of the study was to compare the possible effects of taking tomato puree deriving from hydroponic agriculture versus tomato puree from conventional cultivation in the subjective control of SNAS symptoms.

The secondary outcomes were the evaluation of adherence to hydroponic tomato puree versus conventional tomato puree and the impact on QoL after each planned dietary treatment (hydroponic vs. conventional tomato puree).

The study protocol was reviewed and approved by the Ethics Committee of the Fondazione Policlinico Universitario A. Gemelli IRCCS in Rome, Italy (ID 3643; Prot N.0050336/20). The study was registered on ClinicalTrials.gov (NCT05232890, accessed on 15 September 2022). The study was conducted in adherence to the guidelines of the Declaration of Helsinki and the Consolidated Standards of Reporting Trials (CONSORT).

All patients gave written informed consent.

Figure 1 shows the study design.

### 2.2. Trial Location, Enrollment Strategy, and Study Sample

The study took place in the Allergy Unit of the Fondazione Policlinico Universitario A. Gemelli IRCCS in Rome, Italy, employing clinic-based recruitment (A.R., R.C., G.P., and E.N.).

We screened patients aged 18–65 years with a documented diagnosis of SNAS, defined by the coexistence of: (a) a history of typical skin and gastrointestinal symptoms, (b) a positive nickel sulphate patch test, (c) clinical improvement of at least 70% after 4–6 weeks of a low-Ni diet, and (d) positivity of the oral challenge test with nickel sulphate [14].

Exclusion criteria included: (a) known pregnancy or breastfeeding, (b) organic diseases capable of affecting gastrointestinal symptoms (i.e., celiac disease, poorly controlled diabetes, scleroderma, chronic inflammatory bowel diseases), (c) abuse of coffee, tea, coca cola, and smoking habit. We also excluded patients who have been taking systemic probiotics, antibiotics, or systemic corticosteroids within the past 30 days and patients who have been taking antidepressant or anxiolytic drugs for less than a month. On the other hand, patients could be enrolled if taking the aforementioned drugs at a stable dosage for at least one month. We did not exclude patients with lactose intolerance if the diagnosis was made more than 6 months ago and if the patient did not report adequate symptom relief after at least 6 months of a lactose-free diet.

### 2.3. Randomization, Treatment Assignment, and Masking

The randomization sequence was generated with a 1:1 block randomization (Stata version 9, StataCorp LLC, College Station, TX, USA) by an unblinded staff member (R.I.). The randomization list was known only to an unblinded staff member (R.I.). Two unblinded staff members (A.L. and C.F.) were responsible for labeling the experimental product and the comparison product according to the randomization list. The product was delivered to the patient according to the assigned randomization number. Three unblinded staff members (V.M., C.F., and A.L.) were not involved in any other study procedure and, in particular, did not take part in the evaluation of the outcomes. Clinicians (A.R., R.C., E.N., M.C.M., E.P., A.G., C.R.S., S.P.) and participants were both blinded to group assignment.

The organoleptic characteristics of the two products were similar. The research team asked the co-financier to submit both tomato purees (from hydroponic technology and traditional cultivation) to quantitative analysis of Ni content by an independent Analysis Laboratory, which is registered with Decree of the General Directorate of Health number 893 of 2 February 2011 in the Register of the Lombardy Region and, therefore, a laboratory authorized to carry out analyses in the context of the self-control procedures of the food industries. The investigations revealed a quantity of Ni < 0.015 mg/kg in the hydroponic Datterino tomato puree, while the quantity of Ni in the traditionally derived Datterino tomato sauce was 0.026 mg/kg. The instructions, given to patients regarding the preparation and consumption, were the same for both products.

In the event of any adverse event, the assignment to the intervention for a given participant was revealed to the multidisciplinary team by accessing the assignment list. The principal investigator could have access to the list only in case of an emergency, if knowledge of the treatment type assigned was considered essential to provide the patient with the best possible care.

### 2.4. Trial Interventions

Hydroponic cultivated tomato sauce, derived from drip systems (recovery/non-recovery), was used. In particular, tomato plants grew on the substrate cultivation technique, where their root system grows within the inert substrate rockwool simulating the structure (macro- and micro-porosity) of the soil. Inside this, the nutrient solution, characterized by water and the mineral elements necessary, was administered to meet the needs of the crop.

At the time of enrollment, the patients were randomized to one of the following treatment arms: (1) daily intake of minimum 100 mL to maximum 200 mL of 100% Italian Datterino tomato puree derived from hydroponic (soilless) technology contained in a 720 mL bottle (group A); (2) daily intake of minimum 100 mL to maximum 200 mL of 100% Italian Datterino tomato puree derived from conventional cultivation contained in a 720 mL bottle (group B). This first phase of the study lasted 12 weeks. After this initial treatment phase, both study groups followed a 2-week wash-out period following a low-Ni diet. At the end of the wash-out phase, the two groups crossed over. Therefore, group A was assigned to treatment arm B, and vice versa, for a period of 12 weeks (the second phase of the study).

Furthermore, a nutritional anamnesis was also collected by a team of dieticians (M.C.M. and E.P.) to evaluate the characteristics of the diet followed by the participants and to customize the bromatological composition to balance the overall intake of macro- and micro-nutrients. All enrolled participants underwent anthropometric data collection (weight, height, body mass index, wrist circumference, arm circumference, waist circumference, and hip circumference) and body composition evaluation using a bioimpedance analysis at baseline and last visits of each 12-week trial phase.

Finally, based on the pioneering experience of Braga et al., [13] and the results of our previous work [22], our team of nutritionists elaborated a personalized and balanced dietary plan, depending on personal energy needs and including foods with low-Ni content (Figure 2). This dietary regimen was prescribed to all enrolled patients during the entire study period, also to avoid patients taking the metal from other food sources.

### 2.5. Outcome Measure

#### 2.5.1. Symptoms

The change of symptoms control was assessed by gastroenterologists (C.R.S., S.P., A.G.) using values of a visual analogue scale (VAS) for SNAS before and after taking each type of tomato sauce (hydroponic vs. conventional). The visual analogue scale was modified for the Irritable Bowel Syndrome (IBS) scale [20,28]. This scale included values from a minimum of zero (absence of symptom) to a maximum of ten (highest expression of the symptom). VAS-SNAS data were obtained from all patients at the baseline and last visits of each 12-week trial phase.

#### 2.5.2. Adherence

We collected adherence data through self-compiled daily diary monitoring of tomato puree intake in terms of grams of tomato sauce and open bottles. This strategy allowed us to calculate the actual treatment period, expressed in days, and to obtain as the difference between the number of theoretical days of treatment (84 days) and the number of transient interruptions.

Consequently, adherence to each treatment arm (hydroponic versus conventional tomato puree) was expressed as: (1) the ratio between the actual treatment period and the theoretical days of treatment, (2) the ratio between the actual total amount of grams of tomato sauce and the expected total amount, and (3) the ratio between the actual total number of open bottles and the number of total bottles delivered to the patient. All indices of adherence were reported as a percentage.

#### 2.5.3. Quality of Life

The general QoL and degree of well-being were assessed using the Short-Form 36-Item Health Survey Italian version 2 (SF-36v2) and Psychological General Well-Being Index (PGWBI), respectively.

The SF-36v2 is a self-reported questionnaire comprising 36-items, measuring eight dimensions of general QoL: physical functioning (10 items), role limitation due to physical health problems (4 items), bodily pain (2 items), general health perceptions (5 items), vitality (4 items), social functioning (2 items), role limitations due to emotional problems (3 items), and general mental health (5 items). In addition to individual dimension scores, two summary scores assessing physical and mental dimensions of health and well-being can be calculated, which are the Physical Component Summary (PCS) score and the Mental Component Summary (MCS) score, respectively. Each question score was coded, summed up, and transformed to a scale of 0 (worst possible health state measured by the questionnaire) to 100 (best possible health state) [29].

The PGWBI is a 22-item self-report rating inventory that allows six possible responses for each item. Its score is proportional to the positivity of “well-being” reported during the last 4 weeks, with scores between “0” (the worst condition) and “110” (the best condition). The results are grouped according to well-being levels as: positive well-being (score ≥ 96), no distress (≥73 to ≤95), moderate distress (≥60 to ≤72), and severe distress (≤60). In addition, the scale consists of six domains or dimensions: anxiety (five items; range 0–25), depression (three items; range 0–15), well-being (four items; range 0–20), self-control (three items; range 0–15), general health (three items; range 0–15), and vitality (four items; range 0–20). The response format is graded 1–6 (i.e., total range 22–132), with the highest value corresponding to optimal well-being [30].

All QoL questionnaires were collected from all patients at the baseline and last visits of each 12-week trial phase.

### 2.6. Sample Size

The primary outcome for the crossover trial is symptoms control. For the purpose of calculating sample size, (1) an error estimate of 5%, (2) a chosen confidence level of 80%, (3) an expected mean value of VAS-SNAS of 3.5 after a low-Ni diet and standard deviation (SD) equal to 2.3, as observed in our previous study [20], and (4) an expected reduction of 30% of VAS after a low-Ni diet plus tomato puree derived from hydroponic agriculture compared to a low-Ni diet plus tomato deriving from conventional agriculture, were assumed. Therefore, we estimated a sample of 26 patients (13 per intervention group). Finally, considering a drop-out probability of 10%, we estimated it was necessary to enroll 30 patients (15 for each intervention group).

### 2.7. Statistical Analysis

Continuous data were tested for normal distribution using the Kolmogorov–Smirnov test. Normally distributed and skewed variables were expressed as mean ± standard deviation (SD) and median (interquartile range (IQR = Q3–Q1)), respectively. The categorical data were presented as n (%). The paired Student’s t-test and Mann–Whitney U test were employed for comparisons of outcome measurements between two treatment arms during each phase of the study. The Wilcoxon signed-rank test was used to compare outcome measurements before and after each interventional trial phase. The Bonferroni correction for multiple post hoc comparisons was applied. *p* value < 0.05 was considered significant. Statistical analysis was performed using Stata version 9 (StataCorp LLC, College Station, TX, USA).

## 3. Results

### 3.1. Demographic and Clinical Characteristics

Between February 2021 and March 2022, 60 patients, referred to the Allergy and Clinical Immunology Unit of the Fondazione Policlinico A. Gemelli IRCCS, were assessed for eligibility. Thirty subjects were excluded for the following reasons: 15 patients were already taking tomato derived from hydroponic technology, 4 patients did not fulfil the inclusion criteria, 1 patient was diagnosed with Chron’s disease, and 10 patients were unwilling to participate. The remaining 30 patients were included in the study. Their demographic and clinical characteristics at baseline are detailed in Table 1.

Among the 30 patients randomized to interventional treatments with tomato sauce from hydroponic or conventional agriculture, 8 patients dropped out. Figure 3 describes the CONSORT diagram showing the flow of patients through each stage of the randomized crossover trial.

### 3.2. Symptoms

Daily intake of 100% Italian Datterino tomato puree, deriving from hydroponic technology versus tomato puree deriving from conventional cultivation, significantly reduced gastrointestinal symptoms due to SNAS, except for discomfort, epigastralgia, nausea and itching, in the group A (Table 2). The transition to Datterino tomato puree deriving from conventional cultivation did not change the reported symptoms, except for worsening flatulence (*p* = 0.0577, Wilcoxon signed-rank test).

On the contrary, we did not find any significant differences in the group of nine patients who first received 100% Italian Datterino tomato puree deriving from conventional cultivation, except for the marked improvement in bloating after the shift from tomato puree deriving from conventional cultivation to puree deriving from hydroponic technology (*p* = 0.0060, Wilcoxon signed-rank test). Finally, dermatitis significantly reduced after the low-Ni diet and tomato puree deriving from conventional cultivation (*p* = 0.0084, Wilcoxon signed-rank test).

### 3.3. Adherence

Overall, the participants were very compliant with both interventions, as documented in their food diaries.

The actual treatment period, expressed in days and obtained as the difference between the number of theoretical days of treatment (84 days) and the number of transient interruptions, was 68 days during phase 1 and 63 days during phase 2, respectively, for patients of group A. The actual treatment period of patients of group B was 74 days during phase 1 and 73 days during phase 2, respectively. No differences were found.

Adherence rates to each treatment arm (hydroponic versus conventional tomato puree), expressed as the ratio between the actual treatment period and the theoretical days of treatment, were high in each phase of both interventions (86% during phase 1 and 83% during phase 2, respectively, for patients of group A; 90% during phase 1 and 88% during phase 2, respectively, for patients of group B). No differences were found.

The comparison of treatment adherence, expressed in grams of tomato sauce consumed at the end of each phase for both interventions, did not reveal differences between the two groups. In fact, the median daily amounts of tomato sauce were 148 g (IQR = 178–118.50) during phase 1 and 121 g (IQR = 170–83) during phase 2 for patients of group A (low-Ni diet plus Datterino tomato puree deriving from hydroponic technology and then tomato puree deriving from conventional agriculture), whereas the median daily amounts of tomato sauce were 152 g (IQR = 177.50–133) during phase 1 and 138 g (IQR = 154–134) during phase 2, for patients of group B (low-Ni diet plus Datterino tomato puree deriving from conventional agriculture and then tomato puree deriving from hydroponic technology).

### 3.4. Quality of Life

Overall, the enrolled population had poor QoL prior to the start of the trial as evident from the summary measures of adopted questionnaires (physical component summary, mental component summary, and PGWBI total score). Furthermore, QoL indices of the two intervention groups were substantially homogeneous at baseline (Table 3).

At the end of phase 1, patients randomized to a daily intake of Datterino tomato puree deriving from conventional cultivation (group B) showed significant QoL improvement in three domains of SF-36v2 (physical functioning, *p* = 0.0198; bodily pain, *p* = 0.0258; mental health, *p* = 0.0024) and PGWBI total score (*p* = 0.0105) compared with patients randomized to a daily intake of Datterino tomato puree deriving from hydroponic technology (group A) (Table 4).

On the other hand, patients randomized to a daily intake of Datterino tomato puree deriving from hydroponic technology (group A), after crossover, showed significant QoL improvement in 5 of 10 domains of SF-36v2 (physical functioning, *p* = 0.0000; bodily pain, *p* = 0.0401; vitality, *p* = 0.0034; mental health, *p* = 0.0003; mental component summary, *p* = 0.0141) (Table 4).

## 4. Discussion

In this pilot study, we found that a daily intake of 100% Italian Datterino tomato puree deriving from hydroponic technology associated with a balanced low-Ni diet was a well-appreciated nutritional intervention that significantly reduced gastrointestinal symptoms in an Italian population of patients with SNAS.

A low-Ni diet is a crucial step of the diagnostic process and one of therapeutic pillars for SNAS [13,16,20,21,22,31]. To date, a low-Ni diet has been shown to reduce symptoms and to improve the psycho–physical well-being of SNAS patients [32].

Nevertheless, a long-term low-Ni diet can be restrictive and socially discriminating and is potentially burdened with a high-risk of nutritional deficiencies [33].

Tomato (*Solanum lycopersicum* L.) is most broadly cultivated and consumed worldwide and is a key vegetable not only of the Mediterranean diet; it is the third most important family of commercial crops from an economic point of view [34,35]. Furthermore, tomatoes are known to be naturally rich in Ni, potentially affecting human health [36].

To date, Ni is regulated by European legislation [37] on drinking water with a threshold set at 20 μg L^−1^. Despite the lack of specific legislation on Ni in foods, the European Food Safety Authority (EFSA) [38] defines a tolerable daily dose for body weight of 13 μg Ni kg^−1^.

To enrich the inter-relationship between tomato and Ni, the metal is used as a fertilizer in traditional soil-based agriculture, capable of increasing the quantity and nutritional capacity of the fruit [39], increasing biomass [40], and influencing the content of soluble sugars and L-ascorbic acid, thus reducing the content of acidity, nitrates, and ammonia [41].

Recently, Correia et al., demonstrated that Ni concentration in different parts of the tomato plant (root, stem, leaf, and fruit) increases proportionally with respect to the concentration of the metal in the soil, becoming toxic to the plant [42].

In horticultural practice, the applications of cultures without soil include hydroponic system [43,44].

In our pioneering study, we used Datterino tomatoes derived from soilless hydroponic cultures with rockwool as the mineral substrate.

The primary advantages of rockwool include low bulk density and high overall porosity. Furthermore, rockwool ensures uniform air and water conditions and the distribution of the nutrient solution within the root systems of the plants. Inside this inert substrate, the nutrient solution is administered, characterized by water and the mineral elements necessary to satisfy the needs of the crop. This type of cultivation allows us to know and modulate all the inputs of the production process according to a strict analysis protocol, avoiding unwanted elements [5,45].

To the best of our knowledge, this is the first clinical trial exploring the possible benefits of Datterino tomato puree deriving from hydroponic cultivation in patients with food allergies.

This trial allowed the reintroduction of tomatoes into the diet, a food “desired” by patients with SNAS, relevant from a nutritional point of view and historically used in many recipes of the Mediterranean diet. In fact, the adherence rate for each phase of the study was greater than 80% in both treatment arms. Gladly, most of the patient diaries recorded the comment: “*finally back to eating tomato*”. We can hypothesize that this result is partly attributable to the multidisciplinary team. Starting from our routine clinical practice, we transferred our synergistic and multi-specialist approach into the trial, involving allergologists, gastroenterologists, and nutritionists. Ultimately, the study did not identify the superiority of a treatment with respect to adherence.

Notably, our interventional, randomized, double-blind, single-center crossover controlled clinical trial showed that patients with SNAS reported significant improvement of gastrointestinal symptoms after 12 weeks of daily intake of 100% Italian Datterino tomato puree deriving from hydroponic technology compared with tomato puree deriving from conventional cultivation.

The homogeneity of the two randomized groups concerning the gastrointestinal symptoms reported at baseline, the crossover nature of this study, the intake of a balanced low-Ni diet by both treatment groups, as well as the homogeneous and high adherence to both interventions allow to recognize Datterino tomato puree deriving from hydroponic technology as having a key role in the improvement of most of the gastrointestinal symptoms.

Overall, the enrolled population had poor QoL prior at the start of the trial as evident from indices of adopted questionnaires. This supports previous published data on the significant negative impact of SNAS on psycho–physical well-being [32]. Our study showed that QoL improved after a balanced low-Ni diet combined with Datterino tomato puree deriving from hydroponic technology, without a clear superiority over Datterino tomato puree deriving from conventional cultivation.

This study has some weaknesses. The major criticism could be the drop-out rate: 8 of 30 patients (26%) dropped out of the study. The main reasons given by the patients who dropped out of the study were: (1) the duration of each phase of the study (12 weeks), (2) the negative impact of the COVID-19 pandemic period on treatment adherence, and (3) the summer period included in the overall period of study that limited a daily intake of tomato puree. Another limitation is the lack of a cost-analysis of the intervention, since the use of Datterino tomato puree deriving from hydroponic technology is considered socially expensive for many patients.

Despite these limitations, the crossover nature of the study, the highly selective enrollment criteria, the inter-treatment wash-out period, the multi-disciplinary team, and the personalized nutritional assessment make this study the premise for future trials that conceive *precision medicine* and *precision agriculture* as pillars of management of the multifaceted world of food allergies.

## Figures and Tables

**Figure 1 jcm-11-05459-f001:**
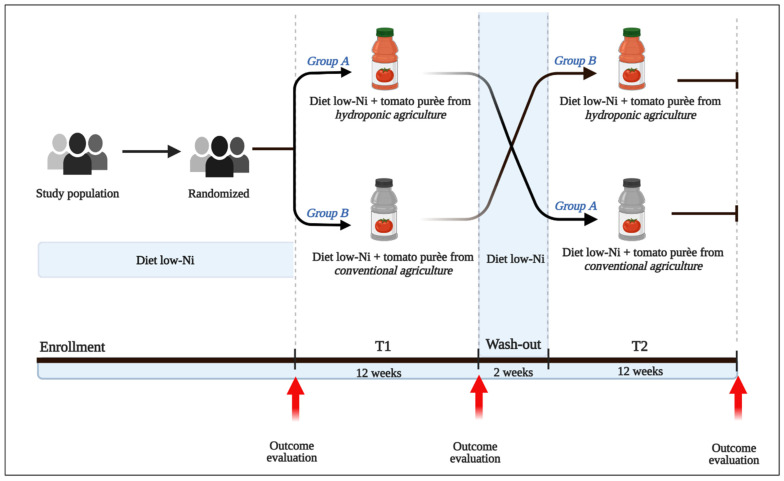
Datterino trial: study design.

**Figure 2 jcm-11-05459-f002:**
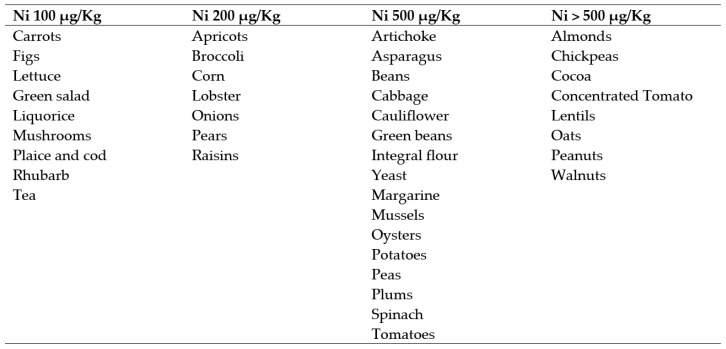
Average amount of Ni in different foods.

**Figure 3 jcm-11-05459-f003:**
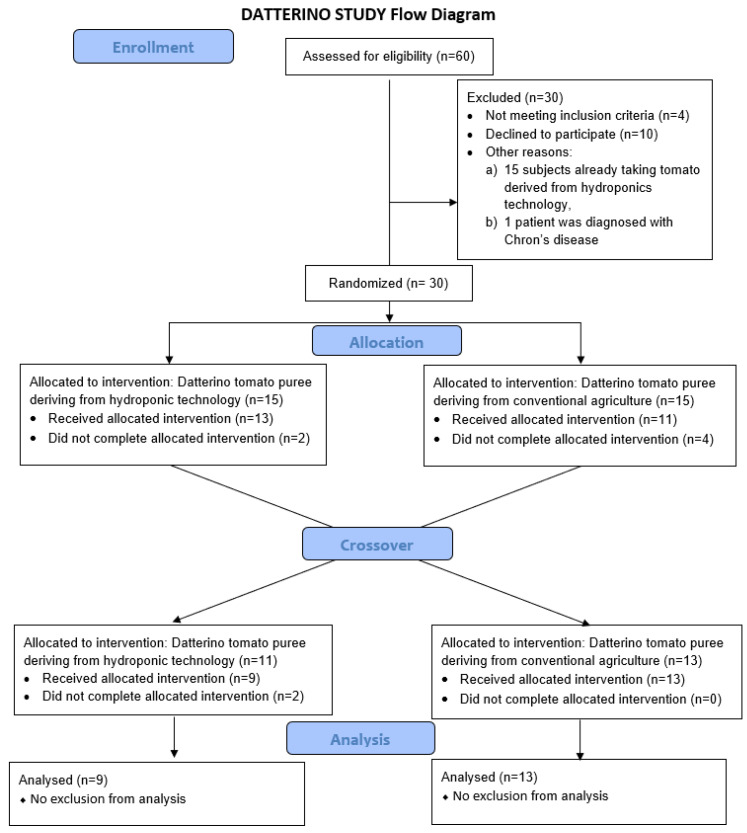
CONSORT diagram flow of patients.

**Table 1 jcm-11-05459-t001:** The baseline socio-demographics and clinical characteristics of the participants.

	Patients (n = 30)					
Variables	N	%	Group A (13 pts)	%	Group B (9 pts)	%
Female gender	29	97	12	92	9	100
Age (mean ± SD ^1^ (range))	39 (±12; 22–61)		42 (±12, 22–61)		38 (±13, 22–56)	
Social class						
Employed	19	63	9	69	6	67
Unemployed	2	7	1	8	0	0
Householder chores	4	13	1	8	1	11
Student	5	17	2	15	2	22
Education						
Less than high school	2	7	1	8	1	11
Short or medium high educ.	9	30	3	23	1	11
University degree	19	63	9	69	7	78
Marital status						
Married	15	50	6	46	4	44
Single	15	50	7	54	5	56
Lactose intolerance	12	40	8	62	2	22
Parental history of allergies	22	73	7	54	9	100
Concomitant other allergies						
known allergies (all)	17		9		5	
Respiratory allergies	13		7		4	
Food allergies	3		2		0	
Adverse drug reaction	7		4		1	
Concomitant positivity to other haptens (palladium chloride, cobalt chloride, potassium dichromate)	17	57	8	62	4	44
Patch grade nickel						
Grade A (±)	0	0	0	0	0	0
Grade B (+)	6	20	2	15	1	11
Grade C (++)	11	37	3	23	3	33
Grade D (+++)	13	43	8	62	5	56

^1^ SD—standard deviation. Group A: 13 patients randomized to Datterino tomato puree deriving from hydroponic technology during phase 1 and Datterino tomato puree deriving from conventional cultivation during phase 2.

**Table 2 jcm-11-05459-t002:** Effects of cross interventions on patient-reported symptoms.

Group A: 13 patients randomized to Datterino tomato puree deriving from hydroponic technology during phase 1 and Datterino tomato puree deriving from conventional cultivation during phase 2					
Symptoms	Baseline M (±SD)	T1 * M (±SD)	T2 ** M (±SD)	*p*-value ^†^ (T1 vs. baseline)	*p*-value ^†^ (T1 vs. T2)
Bloating	6 (±3)	4 (±3)	5 (±3)	0.0111	NS
Discomfort	4 (±4)	2 (±4)	2 (±3)	NS	NS
Flatulence	5 (±3)	3 (±5)	5 (±4)	0.0090	NS
Abdominal cramps	4 (±3)	2 (±2)	1 (±3)	0.0207	NS
Constipation	5 (±4)	3 (±3)	4 (±3)	0.0395	NS
Diarrhea	3 (±3)	2 (±2)	2 (±2)	0.0105	NS
Epigastralgia	4 (±4)	2 (±2)	2 (±3)	NS	NS
Nausea	3 (±3)	2 (±3)	2 (±3)	NS	NS
Vomiting	0	0	0	-	-
Itching	3 (±4)	3 (±3)	2 (±3)	NS	NS
Dermatitis	5 (±4)	3 (±3)	2 (±3)	NS	NS
Group B: 9 patients randomized to Datterino tomato puree deriving from conventional cultivation during phase 1 and Datterino tomato puree deriving from hydroponic technology during phase 2					
Symptoms	Baseline M (±SD)	T1 * M (±SD)	T2 ** M (±SD)	*p*-value ^†^ (T1 vs. baseline)	*p*-value ^†^ (T1 vs. T2)
Bloating	7 (±2)	6 (±3)	3 (±3)	NS	0.0060
Discomfort	4 (±2)	3 (±3)	2 (±3)	NS	NS
Flatulence	4 (±2)	3 (±3)	2 (±3)	NS	NS
Abdominal cramps	4 (±3)	2 (±2)	1 (±1)	NS	NS
Constipation	5 (±4)	4 (±4)	2 (±4)	NS	NS
Diarrhea	3 (±3)	1 (±1)	1 (±3)	NS	NS
Epigastralgia	2 (±2)	2 (±3)	1 (±1)	NS	NS
Nausea	1 (±1)	1 (±2)	0	NS	NS
Vomiting	0	0	0	-	-
Itching	2 (±3)	1 (±1)	1 (±3)	NS	NS
Dermatitis	5 (±4)	1 (±1)	2 (±4)	0.0084	NS

*, T1—end of phase 1. **, T2—end of phase 2. ^†^, comparison between mean values during phase 1 and phase 2 for each symptom (Wilcoxon signed-rank test).

**Table 3 jcm-11-05459-t003:** Results of the QoL questionnaires at baseline.

QoL Questionnaires	ARM 1 * M (±SD)	ARM 2 *** M (±SD)	*p*-Value ^†^ (ARM 1 vs. ARM 2)
SF-36v2 ^~^			
Physical functioning	87 (±20)	97 (±5)	0.0004
Role physical	87 (±19)	81 (±39)	0.0148
Bodily pain	70 (±22)	78 (±20)	NS
General health	54 (±26)	68 (±7)	0.0005
Vitality	52 (±28)	57 (±23)	NS
Social functioning	73 (±20)	82 (±20)	NS
Role emotional	77 (±34)	78 (±44)	NS
Mental health	72 (±23)	74 (±22)	NS
Physical component summary	49 (±7)	53 (±5)	NS
Mental component summary	46 (±12)	50 (±11)	NS
PGWBI ’, total score	58 (±5)	56 (±4)	NS

*, ARM 1 = Group A: 13 patients randomized to Datterino tomato puree deriving from hydroponic technology during phase 1 and Datterino tomato puree deriving from conventional cultivation during phase 2. ARM 2 = Group B: 9 patients randomized to Datterino tomato puree deriving from conventional cultivation during phase 1 and Datterino tomato puree deriving from hydroponic technology during phase 2. ^†^, comparison between mean values of two intervention arms (Two-sample T test). ^~^, SF-36v2—Short-Form 36-Item Health Survey, version 2. ’, PGWBI—Psychological General Well-Being Index.

**Table 4 jcm-11-05459-t004:** Impact of interventional treatments on QoL.

QoL Questionnaires	Group A * (Phase 1) M (±SD ’)	Group B ** (Phase 1) M (± SD)	*p*-Value ^†^ (A versus B, Phase 1)	Group A * (Phase 2) M (±SD)	Group B ** (Phase 2) M (±SD)	*p*-Value ^†^ (A versus B, Phase 2)
SF-36v2 ^~^						
Physical functioning	87 (±15)	97 (±7)	0.0198	86 (±21)	99 (±2)	0.0000
Role physical	87 (±26)	92 (±18)	NS	92 (±19)	89 (±25)	NS
Bodily pain	77 (±22)	86 (±11)	0.0258	79 (±25)	92 (±13)	0.0401
General health	59 (±24)	75 (±15)	NS	57 (±21)	68 (±19)	NS
Vitality	56 (±25)	66 (±19)	NS	60 (±22)	81 (±8)	0.0034
Social functioning	75 (±19)	82 (±14)	NS	81 (±19)	82 (±26)	NS
Role emotional	82 (±35)	85 (±34)	NS	85 (±38)	89 (±24)	NS
Mental health	71 (±22)	83 (±7)	0.0024	70 (±22)	86 (±6)	0.0003
Physical component summary	50 (±5)	52 (±7)	NS	50 (±6)	53 (±5)	NS
Mental component summary	48 (±11)	52 (±7)	NS	50 (±11)	54 (±5)	0.0141

*, Group A: 13 patients randomized to Datterino tomato puree deriving from hydroponic technology during phase 1 and Datterino tomato puree deriving from conventional cultivation during phase 2. **, Group B: 9 patients randomized to Datterino tomato puree deriving from conventional cultivation during phase 1 and Datterino tomato puree deriving from hydroponic technology during phase 2. ^†^, comparison between mean values of two intervention arms at each phase (Two-sample T test, *p* < 0.05). ^~^, SF-36v2—Short-Form 36-Item Health Survey, version 2. ’, SD—Standard Deviation.

## Data Availability

Not applicable.

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
