# Peer review of "Datterino Trial: A Double Blind, Randomized, Controlled, Crossover, Clinical Trial on the Use of Hydroponic Cultivated Tomato Sauce in Systemic Nickel Allergy Syndrome"

_jcm, 2022, doi:10.3390/jcm11185459_

Round 1

Reviewer 1 Report (Previous Reviewer 3)

Author Response

September 5th 2022

To Editor and Reviewers
Journal of Clinical Medicine MDPI

We would like to greatly thank the Editor and Reviewers who encouraged the re-submission of the manuscript.

Please find enclosed the Original article (vers. 2) entitled “DATTERINO Trial: a double blind, randomized, controlled, cross-over, clinical trial on the use of Hydroponic Cultivated Tomato Sauce in Systemic Nickel Allergy Syndrome” by Angela Rizzi, Raffaella Chini, Serena Porcari, Carlo Romano Settanni, Eleonora Persichetti, Vincenzina Mora, Caterina Fanali, Alessia Leonetti, Giuseppe Parrinello, Franziska Michaela Lohmeyer, Riccardo Inchingolo, Maria Cristina Mele, Antonio Gasbarrini and Eleonora Nucera.

In this cover letter, we report point-by-point the details of revisions in the manuscript and our responses to the Reviewers' comments.

Author’s Reply to the Review Report (Reviewer 1)
Comments and Suggestions for Authors

The paper discusses the use of tomato sauce coming from hydroponic cultivated tomatoes for the treatment of systemic nickel allergy. It involves a double blind, randomized, controlled, cross-over, clinical trial using 2 small groups of clinically characterized individuals. The study is just a pilot, with only 22 patients who completed it, however it shows some interesting clinical results and provides promising indications.
We thank the Reviewer for the comment.

For my first revision I pointed out that the main flaw of the study was that the level of Ni consumption, and how this varied among the baseline, the T1, the wash-out and the end, was never measured. Even though in the revised version of the paper the Authors have included the information about the level of Ni in the 2 types of tomato sauce, they are still not discussing whether this difference could be significantly relevant to explain the reduction in the reported symptoms of Group A. The values reported indicate between 2 and 3 times less Ni in the sauce from hydroponic tomatoes compared to the conventional one, nevertheless according to the new table in figure 2 we are still in the range of considered “low-Ni” foods. Moreover, we still have no guesses on how much Ni were these patients consuming before getting enrolled, compared to how much they received during the study. The Authors state at line 437 that “Overall, the enrolled population had poor QoL prior to the start of the trial as evident from indices of adopted questionnaires” and that the study “showed that QoL improved after balanced low-Ni diet combined with Datterino tomato puree deriving from hydroponic technology, without a clear superiority over Datterino tomato puree deriving from conventional cultivation”. If the sole adoption of a low-Ni diet was enough to improve QoL independently from the two groups, we can expect that their diet was at least only moderately low, or not low in Ni before the study. However, the consumption during phase T1 of a low-Ni diet coupled to the sauce from conventional tomatoes that contained only 260 µg of Ni per kg did not significantly affect the reported clinical symptoms in Group B.
We thank the Reviewer for the comment, which allowed us to extend the information regarding the nutritional intervention carried out. All enrolled patients followed a low-nickel diet for at least 4-6 weeks before screening. Furthermore, this nutritional treatment had to result in a clinical improvement of at least 70% for the patient to be enrolled. This improvement is reported as one of the study's inclusion criteria. Therefore, the favourable impact on QoL is attributable to the combined nutritional intervention of Datterino tomato puree deriving from hydroponic technology plus low-Ni diet, although not so marked compared to our expectations.

As already remarked during the first revision, the information about Ni uptake before and during the trial are still missing (no data coming from interviews to the patients to estimate their ordinary Ni uptake, no urine analyses) and generally the discussion part needs to be improved, both taking into account the data generated during the study and previous quantitative data from the literature about Ni intake, body mass and clinical symptoms in allergic individuals.
We thank the Reviewer for the comments. The analysis of nickel on biological samples of the enrolled patients was not foreseen since it was a comparative exploratory study between two tomato purees already available on the market and combined, in a cross way, with the low-Ni diet, nutritional pillar of the treatment of patients with SNAS and already in place prior to the trial. Furthermore, an estimate of the amount of Ni uptake, although potentially deductible taking into account the amounts in grams of tomato puree reported in patients’ diaries, is probably quite approximate considering the low and somewhat variable rate of intestinal absorption of nickel (1- 10%, ref. n° 19). Urinary excretion of metal is also somewhat variable (50-80%, doi: 10.1093/ajcn/66.3.616).
We greatly appreciate the Reviewer's invitation to enrich the discussion section by mentioning published pioneering works focused on the correlations among Ni ingestion, BMI and clinical symptoms in allergic patients, particularly SNAS patients. We believe that these papers will be valuable in the discussion of the expected data from the second phase of our research project, focused on the correlations between the ingestion of tomato puree containing nickel and anthropometric data.

Minor comments:
Materials and Methods: line 169: …Datterino tomato puree, while the quantity of Ni…
Line 204: it should be Table 1, not figure 2. The caption of the table should be changed as “Average amount of Ni in different foods”.
We thank the Reviewer for the comment. We modified accordingly. We preferred to keep the figure as it was modified from previous published works.

References: Ref 34 is still unclear. It should be something like “Production of fruit and vegetables. Apples and tomatoes were the top fruit and vegetable produced in the EU in 2015. Spain, Italy and Poland: main producers. Eurostat, News release 2016”.
We thank the Reviewer for the comment. We modified accordingly (lines 561-562).

With the best regards,

Angela Rizzi, Raffaella Chini, Serena Porcari, Carlo Romano Settanni, Eleonora Persichetti, Vincenzina Mora, Caterina Fanali, Alessia Leonetti, Giuseppe Parrinello, Franziska Michaela Lohmeyer, Riccardo Inchingolo, Maria Cristina Mele, Antonio Gasbarrini and Eleonora Nucera

Corresponding Author:
Riccardo Inchingolo, MD, PhD
UOC Pneumologia, Fondazione Policlinico Universitario A. Gemelli IRCCS. Largo A. Gemelli, 8 – 00168 – Rome, Italy.
riccardo.inchingolo@policlinicogemelli.it

Corresponding Author will receive all editorial communications
The authors declare that the manuscript, or specified parts of it, have not been and will not be submitted elsewhere for publication.

Reviewer 2 Report (New Reviewer)

Major comments

This report seems to be novel in that it integrates medicine and agriculture, and takes into account the regional nature of the diet. However, there are several problems as described below.

First, it is difficult to understand the comparisons in this study because the standards for comparisons change from item to item. Symptom scores were compared between baseline and phase 1, and phases 1 and 2 for each group; on the other hand, QoL indices are compared between groups A and B for each phase. It would be easier to understand the evaluation of symptom scores by comparing groups A and B in the same phase as well as QoL scores.

In addition, the time point of evaluation for the primary outcome was not defined. It would be better to use the evaluation results at the end of phase 1 as the primary outcome and the evaluation results at end of phase 2 as the secondary outcome.

Second, Table 1 shows only the overall patient background. It is necessary to describe the patient background of each group.

Third, originally, 13 patients were needed for each group, making a total of 26 subjects; however, only 22 patients were able to continue the study. This small number of participants may have affected the statistical results. It should be noted in the text that this limitation may have affected the results.

Fourth, the text states "a cohort of SNAS patients following a low-Ni diet for at least 4-6 weeks," however in Figure 1, it appears that the low-Ni diet was first started in this study, not before. It would be better if the description in Figure 1 was also revised.

Additionally, in Table 2, the symptom scores of Group B, who were on a Ni-rich tomato puree diet, also showed a trend toward improvement (rather than exacerbation) at the end of the first period; however, it is possible that the low-Ni diet was not well-managed prior to the study; this point needs to be discussed.

Fifth, the Discussion stated that "tomato puree deriving from hydroponic technology associated with a balanced low-Ni diet significantly reduced gastrointestinal symptoms and improved QoL;" however, this study did not show improvement in QoL. It is necessary to compare QoL scores between baseline and end of phase 1 to show improvement in QoL with tomato puree intake. In addition, Group B always has a higher QoL score than Group A. The authors should discuss this point. Since there is already a difference in the baseline, it would be better to evaluate QoL not only by absolute numbers but also by the percentage change in the scores.

Sixth, in terms of adherence, I understand the "ratio between actual total amount of grams of tomato sauce and expected total amount" and "ratio between actual total number of open bottles and number of total bottles delivered to the patient" are basically the same indicator. Could the authors please explain the difference between these two? In addition, although the text states “Ultimately, the study did not identify superiority of a treatment with respect to adherence,” Figure 4 states that there is a significant difference. The authors should explain this point and what this figure indicates.

Finally, the text defines significance as a p-value <0.05. In Figure 2, however, “Discomfort” and “Dermatitis” in “T1 vs baseline” and “Flatulence” in “T1 vs T2” had p values ≥ 0.05 but were described as having a significant difference. This should be corrected.

Minor comments

Line 360

QoL indices at baseline are not "substantially homogeneous" in the SF-36v2 since there are significant differences in the three items. This should be corrected.

Line 372

In Table 4, the description of the significant differences in 5 out of 10 items for the QoL assessment by SF-36v2 after crossover should be rephrased; the authors should not say that this is "almost" significant.

Line 429

Significant improvement in 5 out of 9 gastrointestinal symptoms is not "almost" significant and should also be reworded.

Line 513

The authors should include the journal and year of publication for Reference 17.

Author Response

September 5th 2022

To Editor and Reviewers
Journal of Clinical Medicine MDPI

We would like to greatly thank the Editor and Reviewers who encouraged the re-submission of the manuscript.

Please find enclosed the Original article (vers. 2) entitled “DATTERINO Trial: a double blind, randomized, controlled, cross-over, clinical trial on the use of Hydroponic Cultivated Tomato Sauce in Systemic Nickel Allergy Syndrome” by Angela Rizzi, Raffaella Chini, Serena Porcari, Carlo Romano Settanni, Eleonora Persichetti, Vincenzina Mora, Caterina Fanali, Alessia Leonetti, Giuseppe Parrinello, Franziska Michaela Lohmeyer, Riccardo Inchingolo, Maria Cristina Mele, Antonio Gasbarrini and Eleonora Nucera.

In this cover letter, we report point-by-point the details of revisions in the manuscript and our responses to the Reviewers' comments.

Author’s Reply to the Review Report (Reviewer 2)
Comments and Suggestions for Authors

Major comments
This report seems to be novel in that it integrates medicine and agriculture, and takes into account the regional nature of the diet. However, there are several problems as described below.
First, it is difficult to understand the comparisons in this study because the standards for comparisons change from item to item. Symptom scores were compared between baseline and phase 1, and phases 1 and 2 for each group; on the other hand, QoL indices are compared between groups A and B for each phase. It would be easier to understand the evaluation of symptom scores by comparing groups A and B in the same phase as well as QoL scores.
We thank the Reviewer for the comment, which allowed us to explain the purposes of our analyses. Given the purely exploratory nature of the study, our primary aim was to evaluate the possible favourable impact of hydroponic tomato puree on symptoms. Therefore, we first assessed the two groups independently, analysing symptom scores at baseline and their changes after each phase of the cross-study, by Wilcoxon signed-rank test.
Subsequently, we compared the QoL scores (secondary outcome) of the two groups at baseline and at the end of each phase since the scores used are generic and not specific for pathology and considering the homogeneity of the two groups in terms of adherence to the treatment, expressed in grams of median daily amounts of tomato sauce. In fact, even if the number of bottles actually consumed at the end of each phase seems to show differences between the two groups (figure 4, analysis required during previous round of review), median daily amounts of tomato sauce, actually consumed during each phase, were similar for both groups.
The homogeneous rate of adherence for both interventions was reported in Discussion section (lines 435-439).

In addition, the time point of evaluation for the primary outcome was not defined. It would be better to use the evaluation results at the end of phase 1 as the primary outcome and the evaluation results at end of phase 2 as the secondary outcome.
We thank the Reviewer for the comment. Given the cross-over design of the study, the primary outcome was evaluated immediately after each the intervention (ClinicalTrials.gov, NCT05232890).

Second, Table 1 shows only the overall patient background. It is necessary to describe the patient background of each group.
We thank the Reviewer for the comment. We modified accordingly (line 292).

Third, originally, 13 patients were needed for each group, making a total of 26 subjects; however, only 22 patients were able to continue the study. This small number of participants may have affected the statistical results. It should be noted in the text that this limitation may have affected the results.
We thank the Reviewer for the comment. This limitation is reported in Discussion section (lines 446-450).

Fourth, the text states "a cohort of SNAS patients following a low-Ni diet for at least 4-6 weeks," however in Figure 1, it appears that the low-Ni diet was first started in this study, not before. It would be better if the description in Figure 1 was also revised.
We thank the Reviewer for the comment. We modified accordingly (lines 129-130).

Additionally, in Table 2, the symptom scores of Group B, who were on a Ni-rich tomato puree diet, also showed a trend toward improvement (rather than exacerbation) at the end of the first period; however, it is possible that the low-Ni diet was not well-managed prior to the study; this point needs to be discussed.
We thank the Reviewer for the comment, which allowed us to extend the information regarding the nutritional intervention carried out. All enrolled patients followed a low-nickel diet for at least 4-6 weeks before screening. Furthermore, this nutritional treatment had to result in a clinical improvement of at least 70% for the patient to be enrolled. This improvement is reported as one of the study's inclusion criteria.

Fifth, the Discussion stated that "tomato puree deriving from hydroponic technology associated with a balanced low-Ni diet significantly reduced gastrointestinal symptoms and improved QoL;" however, this study did not show improvement in QoL. It is necessary to compare QoL scores between baseline and end of phase 1 to show improvement in QoL with tomato puree intake. In addition, Group B always has a higher QoL score than Group A. The authors should discuss this point. Since there is already a difference in the baseline, it would be better to evaluate QoL not only by absolute numbers but also by the percentage change in the scores.
We thank the Reviewer for the comment. We modified accordingly. Regarding the comment on the analysis of the percentage change in the QoL scores, to the best of our knowledge, there are no data in the literature on the minimum significant difference in patients with food allergy, in particular SNAS (doi: 10.1016/j.jclinepi.2020.11.024). Therefore, we prefer to evaluate absolute values.

Sixth, in terms of adherence, I understand the "ratio between actual total amount of grams of tomato sauce and expected total amount" and "ratio between actual total number of open bottles and number of total bottles delivered to the patient" are basically the same indicator. Could the authors please explain the difference between these two? In addition, although the text states “Ultimately, the study did not identify superiority of a treatment with respect to adherence,” Figure 4 states that there is a significant difference. The authors should explain this point and what this figure indicates.
We thank the Reviewer for the comment. We calculated the actual treatment period, and consequently, adopted three indicators of adherence to each treatment arm: 1) ratio between actual treatment period and theoretical days of treatment, 2) ratio between actual total amount of grams of tomato sauce and expected total amount, and 3) ratio between actual total number of open bottles and number of total bottles delivered to the patient. According to the Authors, the most relevant is the second, since it describes the daily amount of tomato sauce actually consumed by the patient, while, the bottle can be opened but not necessarily completely consumed also in relation to the expiry date. Therefore, we prefer to remove Figure 4 if the Reviewer agrees (lines 345-357).

Finally, the text defines significance as a p-value <0.05. In Figure 2, however, “Discomfort” and “Dermatitis” in “T1 vs baseline” and “Flatulence” in “T1 vs T2” had p values ≥ 0.05 but were described as having a significant difference. This should be corrected.
We thank the Reviewer for the comment. We modified Table 2 accordingly (line 319).

Minor comments
Line 360
QoL indices at baseline are not "substantially homogeneous" in the SF-36v2 since there are significant differences in the three items. This should be corrected.
We thank the Reviewer for the comment. QoL indices at baseline are homogenous because the summary measures of adopted questionnaires (PCS and MCS) are not significantly different. Furthermore, Physical functioning and general health domains are greater in the second group (ARM 2), while the other index (role-physical) is greater in the first.

Line 372
In Table 4, the description of the significant differences in 5 out of 10 items for the QoL assessment by SF-36v2 after crossover should be rephrased; the authors should not say that this is "almost" significant.
We thank the Reviewer for the comment. We modified accordingly (line 379).

Line 429
Significant improvement in 5 out of 9 gastrointestinal symptoms is not "almost" significant and should also be reworded.
We thank the Reviewer for the comment. We modified accordingly.

Line 513
The authors should include the journal and year of publication for Reference 17.
We thank the Reviewer for the comment. We modified accordingly.

With the best regards,

Angela Rizzi, Raffaella Chini, Serena Porcari, Carlo Romano Settanni, Eleonora Persichetti, Vincenzina Mora, Caterina Fanali, Alessia Leonetti, Giuseppe Parrinello, Franziska Michaela Lohmeyer, Riccardo Inchingolo, Maria Cristina Mele, Antonio Gasbarrini and Eleonora Nucera

Corresponding Author:
Riccardo Inchingolo, MD, PhD
UOC Pneumologia, Fondazione Policlinico Universitario A. Gemelli IRCCS. Largo A. Gemelli, 8 – 00168 – Rome, Italy.
riccardo.inchingolo@policlinicogemelli.it

Corresponding Author will receive all editorial communications
The authors declare that the manuscript, or specified parts of it, have not been and will not be submitted elsewhere for publication.

Round 2

Reviewer 2 Report (New Reviewer)

No further change should be required.

This manuscript is a resubmission of an earlier submission. The following is a list of the peer review reports and author responses from that submission.

Round 1

Reviewer 1 Report

The manuscript of Angela Rizzi and co-authors is devoted to the study of the possibility of eating hydroponic cultivated tomatoes in patients with systemic nickel allergy syndrome (SNAS). The problem of SNAS is certainly relevant today. And it is known that the only way to reduce allergy symptoms is a low-nickel diet. However, the scientific significance of this study is not entirely clear. There are a number of questions and comments.

- The article completely lacks information on the analysis of the nickel content in tomatoes (or tomato puree) that were eaten by patients, both conventionally grown and hydroponically grown (only general literature data). If tomatoes from the hydroponic system contained less nickel, then patients who consumed them received less nickel. It is absolutely natural that at the same time their health was better than people who ate food with a high content of nickel.

- There are no data on the third control group of patients who were on the same diet and did not consume tomatoes.

- There is no description of the low-nickel diet. It is not clear whether the content and bioavailability (absorption) of nickel was the same for all patients.

- The meaning and conclusions from the crossover experiment is not entirely clear (Figure 1 contains an error? group A received conventionally grown tomatoes for the second 12 weeks and group B vice versa?).

- Patients consumed 100-200 ml of tomato puree. But this is twice as much, and there is no data on the nickel content in the diet and consumed tomato puree.

- Although the experiment was quite long, there are no data on biochemical indicators of the general condition and allergic status of patients (questionnaire only).

Author Response

July, 10th 2022

To Editor and Reviewers
Nutrients MDPI

We would like to greatly thank the Editor and Reviewers who encouraged a revision of the manuscript.

Please find the enclosed the Revision vers. 1 of the Original article entitled “DATTERINO Trial: a double blind, randomized, controlled, cross-over, clinical trial on the use of Hydroponic Cultivated Tomato Sauce in Systemic Nickel Allergy Syndrome” by Angela Rizzi, Raffaella Chini, Serena Porcari, Carlo Romano Settanni, Eleonora Persichetti, Vincenzina Mora, Caterina Fanali, Alessia Leonetti, Giuseppe Parrinello, Franziska Michaela Lohmeyer, Riccardo Inchingolo, Maria Cristina Mele, Antonio Gasbarrini and Eleonora Nucera.

[Journal of Clinical Medicine] Manuscript ID: jcm-1784715- Major Revisions

Author's Reply to the Review Report (Reviewer 1)
Comments and Suggestions for Authors

The manuscript of Angela Rizzi and co-authors is devoted to the study of the possibility of eating hydroponic cultivated tomatoes in patients with systemic nickel allergy syndrome (SNAS). The problem of SNAS is certainly relevant today. And it is known that the only way to reduce allergy symptoms is a low-nickel diet. However, the scientific significance of this study is not entirely clear. There are a number of questions and comments.

The article completely lacks information on the analysis of the nickel content in tomatoes (or tomato puree) that were eaten by patients, both conventionally grown and hydroponically grown (only general literature data). If tomatoes from the hydroponic system contained less nickel, then patients who consumed them received less nickel. It is absolutely natural that at the same time their health was better than people who ate food with a high content of nickel.
We thank the Reviewer for the comment. We modified the manuscript according to Reviewer’s comments. In particular, we described the results of analysis of Ni quantity provided by an independent Analysis Laboratory, registered, with Decree of the General Directorate of Health number 893 of February 2, 2011, in the Register of the Lombardy Region, a laboratory authorized to carry out analyses in the context of the self-control procedures of the food industries. Lines: 162-170.

There are no data on the third control group of patients who were on the same diet and did not consume tomatoes.
We thank the Reviewer for the comment. This comparison was not foreseen in the study design, since low-Ni diet is an approved nutritional treatment for these patients and, moreover, the two tomato purees are already available on the market. Therefore, the experimental design concerned the direct comparison between the two purees.

There is no description of the low-nickel diet. It is not clear whether the content and bioavailability (absorption) of nickel was the same for all patients.
We thank the Reviewer for the comment. We modified the “Trial interventions” paragraph adding details on low-Ni diet and a dedicated figure (Figure 2) describing different Ni contents in groups of foods. Lines: 202-207.

The meaning and conclusions from the crossover experiment is not entirely clear (Figure 1 contains an error? group A received conventionally grown tomatoes for the second 12 weeks and group B vice versa?).
We thank the Reviewer for the comment. This was a typo, we modified the Figure 1 accordingly. Lines: 130-131.

Patients consumed 100-200 ml of tomato puree. But, this is twice as much, and there is no data on the nickel content in the diet and consumed tomato puree.
We thank the Reviewer for the comment. We modified the “Trial interventions” paragraph adding details on low-Ni diet and a dedicated figure (Figure 2) describing different Ni contents in groups of foods (lines: 162-170). Furthermore, we specified the comparison of treatment adherence, expressed in grams of tomato sauce, actually consumed at the end of each phase for both interventions. Lines: 329-338. We can add a dedicated figure if appreciated by the Reviewer.

Although the experiment was quite long, there are no data on biochemical indicators of the general condition and allergic status of patients (questionnaire only).
We thank the Reviewer for the comment. We collected biochemical indicators as part of this research project, but we focused our attention first on symptoms’ control in order to explore if this nutritional intervention improved well-being of this population. Allergic status of patients was described in Table 1.

With the best regards,

Angela Rizzi, Raffaella Chini, Serena Porcari, Carlo Romano Settanni, Eleonora Persichetti, Vincenzina Mora, Caterina Fanali, Alessia Leonetti, Giuseppe Parrinello, Franziska Michaela Lohmeyer, Riccardo Inchingolo, Maria Cristina Mele, Antonio Gasbarrini and Eleonora Nucera

Corresponding Author:
Riccardo Inchingolo, MD, PhD
UOC Pneumologia, Fondazione Policlinico Universitario A. Gemelli IRCCS. Largo A. Gemelli, 8 – 00168 – Rome, Italy.
riccardo.inchingolo@policlinicogemelli.it

Corresponding Author will receive all editorial communications
The authors declare that the manuscript, or specified parts of it, have not been and will not be submitted elsewhere for publication.

Reviewer 2 Report

Rizzi et al. conducted a clinical trial to prove the benefit of consuming hydroponic foods (tomato) in systemic nickel allergy syndrome (SNAS). Although this syndrome has a limited approach in countries other than Italy, the bibliography supports the recognition of this entity. The findings are interesting however, some concerns are listed below.

SNAS is a disease with little recognition in other regions of the world, thus, the definitions should be supported by indexed bibliography, for example, reference #17 cannot be found on the web (Falagiani, P.; Gioacchino, M.D.; Ricciardi, L.; Minciullo, P.L.; Saitta, S.; Carní, A.; Santoro, G.; Gangemi, S.; Minelli, 499 M.; Bozzetti, M.P.; et al. Systemic Nickel Allergy Syndrome (SNAS): A Review. 13).

The small sample size is well justified because the calculation was based on earlier bibliography; however, parametric statistics should not be implemented in all analyses, both in descriptive and bivariate statistics. Some values could be presented with interquartile range and median. Similarly, the paired analysis in two samples could have been better analyzed with the Wilcoxon test; particularly in the data presented in Table 2 (patient-reported symptoms).

The results obtained in the quality of life showed a significant modification in different domains of the validated questionnaire SF-36v2, however, figure 5 is not representative of these modifications. Since the data are paired, the best way to show data is via before after graph. The authors should consider a better way to present the remarkable results in a more conventional graph and should add a table with the significant and non-significant results in the measurement of quality of life.

Author Response

July, 10th 2022

To Editor and Reviewers
Nutrients MDPI

We would like to greatly thank the Editor and Reviewers who encouraged a revision of the manuscript.

Please find the enclosed the Revision vers. 1 of the Original article entitled “DATTERINO Trial: a double blind, randomized, controlled, cross-over, clinical trial on the use of Hydroponic Cultivated Tomato Sauce in Systemic Nickel Allergy Syndrome” by Angela Rizzi, Raffaella Chini, Serena Porcari, Carlo Romano Settanni, Eleonora Persichetti, Vincenzina Mora, Caterina Fanali, Alessia Leonetti, Giuseppe Parrinello, Franziska Michaela Lohmeyer, Riccardo Inchingolo, Maria Cristina Mele, Antonio Gasbarrini and Eleonora Nucera.

[Journal of Clinical Medicine] Manuscript ID: jcm-1784715- Major Revisions

Author's Reply to the Review Report (Reviewer 2)
Comments and Suggestions for Authors

Rizzi et al. conducted a clinical trial to prove the benefit of consuming hydroponic foods (tomato) in systemic nickel allergy syndrome (SNAS). Although this syndrome has a limited approach in countries other than Italy, the bibliography supports the recognition of this entity. The findings are interesting however, some concerns are listed below.

SNAS is a disease with little recognition in other regions of the world, thus, the definitions should be supported by indexed bibliography, for example, reference #17 cannot be found on the web (Falagiani, P.; Gioacchino, M.D.; Ricciardi, L.; Minciullo, P.L.; Saitta, S.; Carní, A.; Santoro, G.; Gangemi, S.; Minelli, 499 M.; Bozzetti, M.P.; et al. Systemic Nickel Allergy Syndrome (SNAS): A Review. 13).
We thank the Reviewer for the comment. We changed the reference n° 17.

The small sample size is well justified because the calculation was based on earlier bibliography; however, parametric statistics should not be implemented in all analyses, both in descriptive and bivariate statistics. Some values could be presented with interquartile range and median. Similarly, the paired analysis in two samples could have been better analyzed with the Wilcoxon test; particularly in the data presented in Table 2 (patient-reported symptoms).
We thank the Reviewer for the comment. As reported in “Statistical Analysis” section, we assessed if the distribution of the outcome variables differed significantly from normality by the Kolmogorov-Smirnov statistical test. The hypothesis, defined a priori, was the superiority of the variable of interest in the group of patients candidates to Datterino tomato puree in phase 1 and phase 2. Except for the non-normal distribution of treatment adherence, expressed in grams of tomato puree consumed at the end of each phase for both interventions, all other variables showed normal distribution. Therefore, even if the sample analysed was small, we applied the a priori analysis by comparing the values of continuous variables between the two groups using the Mann-Whitney U-test. We modified “Adherence” paragraph accordingly. As regards table 2 (patient-reported symptoms), we used paired Student’s t test since variables were normally distributed.

The results obtained in the quality of life showed a significant modification in different domains of the validated questionnaire SF-36v2, however, figure 5 is not representative of these modifications. Since the data are paired, the best way to show data is via before after graph. The authors should consider a better way to present the remarkable results in a more conventional graph and should add a table with the significant and non-significant results in the measurement of quality of life.
We thank the Reviewer for the comment. We added a dedicated table (Table 4) and removed figure 5.

With the best regards,

Angela Rizzi, Raffaella Chini, Serena Porcari, Carlo Romano Settanni, Eleonora Persichetti, Vincenzina Mora, Caterina Fanali, Alessia Leonetti, Giuseppe Parrinello, Franziska Michaela Lohmeyer, Riccardo Inchingolo, Maria Cristina Mele, Antonio Gasbarrini and Eleonora Nucera

Corresponding Author:
Riccardo Inchingolo, MD, PhD
UOC Pneumologia, Fondazione Policlinico Universitario A. Gemelli IRCCS. Largo A. Gemelli, 8 – 00168 – Rome, Italy.
riccardo.inchingolo@policlinicogemelli.it

Corresponding Author will receive all editorial communications

The authors declare that the manuscript, or specified parts of it, have not been and will not be submitted elsewhere for publication.

Reviewer 3 Report

The paper discusses the use of tomato sauce coming from hydroponic cultivated tomatoes for the treatment of systemic nickel allergy. It involves a double blind, randomized, controlled, cross-over, clinical trial using 2 small groups of clinically characterized individuals. The study is just a pilot, with only 22 patients who completed it, however it shows some interesting clinical results and provides promising indications.

From my perspective the main flaw of the study is that the level of Ni consumption, and how this varies among the baseline, the T1, the wash-out and the end, is always assumed, never measured, and not even guessed. From one side the Authors state in the introduction that the Ni content in the food is highly variable, to the extent that is difficult to follow a nutritional approach based on a low-Ni diet, and from the other side the levels of Ni in the 2 types of tomato sauce are not measured.  The clinical profile of the Ni-allergic individuals included in the study is well characterized: most of the participants have medium to strong symptoms, so, even though they do not regularly consume hydroponic tomatoes, one can assume that due to their condition their normal diet tends to be low on Ni. However, this information is missing, and there are not even other data supporting their common Ni uptake, for example urine analyses. When the study started, both groups received a low-Ni diet, and this seems to have an impact on their reported symptoms, independently whether they are in group A or B. This means that the diet they followed before the study was anyway not so strict as the one they receive during the study, which is in line with what reported in the discussion at lines 370-371. However, it would nice if this was a bit more grounded with analytical data.

Minor comments:

Abstract: line 36: assigned to one OF the following

Lines 37-39: weekS

Introduction: lines 65-66: less time to grow and faster growth is the same thing.

Lines 73-74: “The average concentration of Ni is 500 ug/kg in fresh tomatoes, and higher in concentrated ones”. How much should we expect instead from the hydroponic culture? This information is relevant to put the 2 diets in perspective.

Lines 87-88: Reference 19 is too outdated, especially since the sentence starts with “A modern diet”.

Line 89: How much Ni are we talking about when we speak of a low-Ni diet? Is it more or less than the diet received by the patients during the wash-out?

Line 104: obtainING

Materials and Methods: line 183: weekS

Discussion: line 375: what is reference 34?

Line 419: …low-Ni diet by both treatment groups, AS WELL AS the homogeneous and high adherence…

Figure 2: in the Allocation section, the number between parenthesis on the right after “Received allocated intervention” is 11 and not 15.

Figure 3: introduce a dashed line between patients 30 and 9 to better visualize the 2 groups.

Author Response

July, 10th 2022

To Editor and Reviewers
Journal of Clinical Medicine MDPI

We would like to greatly thank the Editor and Reviewers who encouraged a revision of the manuscript.

Please find the enclosed the Revision vers. 1 of the Original article entitled “DATTERINO Trial: a double blind, randomized, controlled, cross-over, clinical trial on the use of Hydroponic Cultivated Tomato Sauce in Systemic Nickel Allergy Syndrome” by Angela Rizzi, Raffaella Chini, Serena Porcari, Carlo Romano Settanni, Eleonora Persichetti, Vincenzina Mora, Caterina Fanali, Alessia Leonetti, Giuseppe Parrinello, Franziska Michaela Lohmeyer, Riccardo Inchingolo, Maria Cristina Mele, Antonio Gasbarrini and Eleonora Nucera.

[Journal of Clinical Medicine] Manuscript ID: jcm-1784715- Major Revisions

Author's Reply to the Review Report (Reviewer 3)
Comments and Suggestions for Authors

The paper discusses the use of tomato sauce coming from hydroponic cultivated tomatoes for the treatment of systemic nickel allergy. It involves a double blind, randomized, controlled, cross-over, clinical trial using 2 small groups of clinically characterized individuals. The study is just a pilot, with only 22 patients who completed it, however it shows some interesting clinical results and provides promising indications.

From my perspective the main flaw of the study is that the level of Ni consumption, and how this varies among the baseline, the T1, the wash-out and the end, is always assumed, never measured, and not even guessed. From one side the Authors state in the introduction that the Ni content in the food is highly variable, to the extent that is difficult to follow a nutritional approach based on a low-Ni diet, and from the other side the levels of Ni in the 2 types of tomato sauce are not measured.  The clinical profile of the Ni-allergic individuals included in the study is well characterized: most of the participants have medium to strong symptoms, so, even though they do not regularly consume hydroponic tomatoes, one can assume that due to their condition their normal diet tends to be low on Ni. However, this information is missing, and there are not even other data supporting their common Ni uptake, for example urine analyses. When the study started, both groups received a low-Ni diet, and this seems to have an impact on their reported symptoms, independently whether they are in group A or B. This means that the diet they followed before the study was anyway not so strict as the one they receive during the study, which is in line with what reported in the discussion at lines 370-371. However, it would nice if this was a bit more grounded with analytical data.
We thank the Reviewer for the comments, which allowed us to extend the information regarding the nutritional intervention carried out. In particular, we reported Ni content present in tomato puree before the start of the trial and we specified the average Ni content in foods (see figure 2). The analysis of nickel on biological samples of the enrolled patients was not foreseen since it was a comparative exploratory study between two tomato purees already available on the market and combined, in a cross way, with the low-Ni diet, nutritional pillar of the treatment of patients with SNAS and already in place prior to the trial (see inclusion criteria). We modified the manuscript accordingly. Lines: 162-170 and 202-207.

Minor comments:

Abstract: line 36: assigned to one OF the following.
We thank the Reviewer for the comment. We modified accordingly.

Lines 37-39: weekS.
We thank the Reviewer for the comment. We modified accordingly.

Introduction: lines 65-66: less time to grow and faster growth is the same thing.
We thank the Reviewer for the comment. We modified accordingly.

Lines 73-74: “The average concentration of Ni is 500 ug/kg in fresh tomatoes, and higher in concentrated ones”. How much should we expect instead from the hydroponic culture? This information is relevant to put the 2 diets in perspective.
We thank the Reviewer for the comment. It is known that hydroponics allows controlled production of vegetables free from harmful substances, as reported in lines 100-104. We added the results of the analysis of Ni quantity provided by an independent Analysis Laboratory, registered, with Decree of the General Directorate of Health number 893 of February 2, 2011, in the Register of the Lombardy Region, a laboratory authorized to carry out analyses in the context of the self-control procedures of the food industries. Lines: 162-170.

Lines 87-88: Reference 19 is too outdated, especially since the sentence starts with “A modern diet”.
We thank the Reviewer for the comment. We replaced “modern” with “common”. Line 86.

Line 89: How much Ni are we talking about when we speak of a low-Ni diet? Is it more or less than the diet received by the patients during the wash-out?
We thank the Reviewer for the comment. The diet is the same. Low-Ni diet is: 1) one of the inclusion criteria (clinical improvement of at least 70% after 4-6 weeks), 2) the basic balanced diet taken by patients during the two treatment phases and 3) the diet taken during the wash-out period.

Line 104: obtainING
We thank the Reviewer for the comment. We modified accordingly.

Materials and Methods: line 183: weekS
We thank the Reviewer for the comment. We modified accordingly.

Discussion: line 375: what is reference 34?
We thank the Reviewer for the comment. We modified accordingly.

Line 419: …low-Ni diet by both treatment groups, AS WELL AS the homogeneous and high adherence…
We thank the Reviewer for the comment. We modified accordingly.

Figure 2: in the Allocation section, the number between parenthesis on the right after “Received allocated intervention” is 11 and not 15.
We thank the Reviewer for the comment. We modified accordingly.

Figure 3: introduce a dashed line between patients 30 and 9 to better visualize the 2 groups.
We thank the Reviewer for the comment. We modified accordingly.

With the best regards,
Angela Rizzi, Raffaella Chini, Serena Porcari, Carlo Romano Settanni, Eleonora Persichetti, Vincenzina Mora, Caterina Fanali, Alessia Leonetti, Giuseppe Parrinello, Franziska Michaela Lohmeyer, Riccardo Inchingolo, Maria Cristina Mele, Antonio Gasbarrini and Eleonora Nucera

Corresponding Author:
Riccardo Inchingolo, MD, PhD
UOC Pneumologia, Fondazione Policlinico Universitario A. Gemelli IRCCS. Largo A. Gemelli, 8 – 00168 – Rome, Italy.
riccardo.inchingolo@policlinicogemelli.it

Corresponding Author will receive all editorial communications
The authors declare that the manuscript, or specified parts of it, have not been and will not be submitted elsewhere for publication.

Reviewer 4 Report

This study is aimed to compare the effects of one tomato puree preparation deriving from hydroponic agriculture vs. one tomato puree preparation from a conventional cultivation. As it seems that major discussed claim would indicate that observed differences are consequent to the specificity of hydroponic technique, no real evidences are provided to support this conclusion. It is reported that "organoleptic characteristics of the two products were similar", but no chemical analysis data of the two food preparations was performed, not even to assess Ni concentration. It seems that both tomato preparations improve patient status, even with significative differences, although only in some cases, but it is difficult to evaluate the specific effect of low Ni diet alone, which might also be even preponderant. Therefore this manuscript is not publishable as much more data are necessary to support possible conclusions: more than single preparations should be considered, chemically analyzed and compared. In addition, experimental design should allow to distinguish the specific effect of low Ni diet from that of the tomato preparations, while patients number, being quite limited, should be increased. Manuscript indicate comments about a fig.5 which is not present, possibly to be referred to fig.4. In addition to data referred to patients groups, it could also be interesting to report individual data from each patient.

Author Response

July, 10th 2022

To Editor and Reviewers
Journal of Clinical Medicine MDPI

We would like to greatly thank the Editor and Reviewers who encouraged a revision of the manuscript.

Please find the enclosed the Revision vers. 1 of the Original article entitled “DATTERINO Trial: a double blind, randomized, controlled, cross-over, clinical trial on the use of Hydroponic Cultivated Tomato Sauce in Systemic Nickel Allergy Syndrome” by Angela Rizzi, Raffaella Chini, Serena Porcari, Carlo Romano Settanni, Eleonora Persichetti, Vincenzina Mora, Caterina Fanali, Alessia Leonetti, Giuseppe Parrinello, Franziska Michaela Lohmeyer, Riccardo Inchingolo, Maria Cristina Mele, Antonio Gasbarrini and Eleonora Nucera.

[Journal of Clinical Medicine] Manuscript ID: jcm-1784715- Major Revisions

Author's Reply to the Review Report (Reviewer 4)
Comments and Suggestions for Authors

This study is aimed to compare the effects of one tomato puree preparation deriving from hydroponic agriculture vs. one tomato puree preparation from a conventional cultivation. As it seems that major discussed claim would indicate that observed differences are consequent to the specificity of hydroponic technique, no real evidences are provided to support this conclusion. It is reported that "organoleptic characteristics of the two products were similar", but no chemical analysis data of the two food preparations was performed, not even to assess Ni concentration. It seems that both tomato preparations improve patient status, even with significative differences, although only in some cases, but it is difficult to evaluate the specific effect of low Ni diet alone, which might also be even preponderant. Therefore this manuscript is not publishable as much more data are necessary to support possible conclusions: more than single preparations should be considered, chemically analysed and compared. In addition, experimental design should allow to distinguish the specific effect of low Ni diet from that of the tomato preparations, while patients number, being quite limited, should be increased. Manuscript indicate comments about a fig.5 which is not present, possibly to be referred to fig.4. In addition to data referred to patients groups, it could also be interesting to report individual data from each patient.
We thank the Reviewer for the comments, which allowed us to extend the information regarding the nutritional intervention carried out. In particular, we reported Ni content present in tomato puree before the start of the trial and we specified the average Ni content in foods (see figure 2). The results of the analysis of Ni quantity were provided by an independent Analysis Laboratory, registered, with Decree of the General Directorate of Health number 893 of February 2, 2011, in the Register of the Lombardy Region, a laboratory authorized to carry out analyses in the context of the self-control procedures of the food industries. Lines: 162-170.
As concerns the impact of the low-Ni diet alone, this was not foreseen in the study design, since low-Ni diet is an approved nutritional treatment for these patients and, moreover, the two tomato purees are already available on the market. Therefore, the experimental design concerned the direct comparison between the two tomato purees. Regarding the number of patients, considered as limited by the Reviewer, it satisfies the sample size calculated a priori. Therefore, we respect but do not share the comment. We thank the Reviewer for the comment on Figure 5, removed after adding a new table. Given the number of data and the length of the manuscript, as evidenced by the other Reviewers, we prefer not to further extend the manuscript with other data for individual patients.

With the best regards,

Angela Rizzi, Raffaella Chini, Serena Porcari, Carlo Romano Settanni, Eleonora Persichetti, Vincenzina Mora, Caterina Fanali, Alessia Leonetti, Giuseppe Parrinello, Franziska Michaela Lohmeyer, Riccardo Inchingolo, Maria Cristina Mele, Antonio Gasbarrini and Eleonora Nucera

Corresponding Author:
Riccardo Inchingolo, MD, PhD
UOC Pneumologia, Fondazione Policlinico Universitario A. Gemelli IRCCS. Largo A. Gemelli, 8 – 00168 – Rome, Italy.
riccardo.inchingolo@policlinicogemelli.it

Corresponding Author will receive all editorial communications
The authors declare that the manuscript, or specified parts of it, have not been and will not be submitted elsewhere for publication.

Round 2

Reviewer 1 Report

The authors answered most of the questions and added some new information to the manuscript. But, unfortunately, in my opinion, the manuscript cannot be published in this form.

It is still unclear what exactly constituted a low-nickel diet. Did all patients receive the same amount of nickel per day? The difference in nickel content in tomatoes grown conventionally and hydroponically differed by less than two times. At the same time, the consumption of tomato puree by patients ranged from 100-200 ml. There are no data for the control group, which did not receive puree, but was on a low-nickel diet. At the same time, the difference in the health of patients from the two groups (questionnaires, Table 4) is not so significant. There are no at least some biochemical data of patients (only questionnaire data about their feeling). All this together does not provide strong evidence in favor of the authors' hypothesis.

Author Response

July, 25th 2022

To Editor and Reviewers
Journal of Clinical Medicine MDPI

We would like to greatly thank the Editor and Reviewers who encouraged a further revision of the manuscript.

Please find the enclosed the Revision vers. 2 of the Original article entitled “DATTERINO Trial: a double blind, randomized, controlled, cross-over, clinical trial on the use of Hydroponic Cultivated Tomato Sauce in Systemic Nickel Allergy Syndrome” by Angela Rizzi, Raffaella Chini, Serena Porcari, Carlo Romano Settanni, Eleonora Persichetti, Vincenzina Mora, Caterina Fanali, Alessia Leonetti, Giuseppe Parrinello, Franziska Michaela Lohmeyer, Riccardo Inchingolo, Maria Cristina Mele, Antonio Gasbarrini and Eleonora Nucera.

[Journal of Clinical Medicine] Manuscript ID: jcm-1784715- Major Revisions

Author’s Reply to the Review Report (Reviewer 1)
Comments and Suggestions for Authors

The authors answered most of the questions and added some new information to the manuscript. But, unfortunately, in my opinion, the manuscript cannot be published in this form.
It is still unclear what exactly constituted a low-nickel diet. Did all patients receive the same amount of nickel per day? The difference in nickel content in tomatoes grown conventionally and hydroponically differed by less than two times. At the same time, the consumption of tomato puree by patients ranged from 100-200 ml. There are no data for the control group, which did not receive puree, but was on a low-nickel diet. At the same time, the difference in the health of patients from the two groups (questionnaires, Table 4) is not so significant. There are no at least some biochemical data of patients (only questionnaire data about their feeling). All this together does not provide strong evidence in favor of the authors’ hypothesis.
We thank the Reviewer for the comment, which allowed us to detail the underlying nutritional treatment of the study. We specified the low-Ni diet prescribed based on pioneering work of Braga et al. [ref. 13 in the manuscript] and results of our previous experience [ref. 22 in the manuscript]. This consolidated dietary treatment is one of the few cornerstones of the treatment of SNAS patients. Therefore, we didn’t design a comparison among the two tomato sauces and low-Ni diet in allocation ratio 1:1:1, but we compared the two tomato sauces, with low-Ni diet as underlying nutritional treatment of the study in order to avoid patients taking the metal from other food sources. We modified the manuscript accordingly (lines: 202-206).
As regards the consumption of tomato sauce, the results obtained from the daily diaries showed a substantially overlapping consumption between the two groups as reported in the manuscript (lines 332-341). The range defined a priori (100-200ml) derives from the estimate of the need for micronutrients in the definition of the personalized diet. The lack of differences between the quantities of tomato puree taken both daily and weekly makes the two groups substantially homogeneous and reinforces the favourable impact of hydroponic tomato puree on symptom control.
Regarding the comment on the impact of nutritional interventions on QoL, to the best of our knowledge, there are no data in the literature on the minimum significant difference in patients with food allergy, in particular SNAS (doi: 10.1016/j.jclinepi.2020.11.024). Therefore, we do not accept the Reviewer's comment.
Finally, we collected biochemical indicators as part of this research project, but we focused our attention first on symptoms’ control in order to explore if this nutritional intervention improved well-being of this population.

With the best regards,

Angela Rizzi, Raffaella Chini, Serena Porcari, Carlo Romano Settanni, Eleonora Persichetti, Vincenzina Mora, Caterina Fanali, Alessia Leonetti, Giuseppe Parrinello, Franziska Michaela Lohmeyer, Riccardo Inchingolo, Maria Cristina Mele, Antonio Gasbarrini and Eleonora Nucera

Corresponding Author:
Riccardo Inchingolo, MD, PhD
UOC Pneumologia, Fondazione Policlinico Universitario A. Gemelli IRCCS. Largo A. Gemelli, 8 – 00168 – Rome, Italy.
riccardo.inchingolo@policlinicogemelli.it

Corresponding Author will receive all editorial communications
The authors declare that the manuscript, or specified parts of it, have not been and will not be submitted elsewhere for publication.

Reviewer 2 Report

The manuscript can now be accepted for publication.

Author Response

July, 25th 2022

To Editor and Reviewers
Journal of Clinical Medicine MDPI

Author's Reply to the Review Report (Reviewer 2)
Comments and Suggestions for Authors

The manuscript can now be accepted for publication.
We thank the Reviewer for the comment.

With the best regards,

Angela Rizzi, Raffaella Chini, Serena Porcari, Carlo Romano Settanni, Eleonora Persichetti, Vincenzina Mora, Caterina Fanali, Alessia Leonetti, Giuseppe Parrinello, Franziska Michaela Lohmeyer, Riccardo Inchingolo, Maria Cristina Mele, Antonio Gasbarrini and Eleonora Nucera

Corresponding Author:
Riccardo Inchingolo, MD, PhD
UOC Pneumologia, Fondazione Policlinico Universitario A. Gemelli IRCCS. Largo A. Gemelli, 8 – 00168 – Rome, Italy.
riccardo.inchingolo@policlinicogemelli.it

Corresponding Author will receive all editorial communications
The authors declare that the manuscript, or specified parts of it, have not been and will not be submitted elsewhere for publication.

Reviewer 4 Report

While the addition of data about Ni content is important, this does not change the main point impeding the publication of this paper: results are far to be sufficient to conclude that the use of hydroponic cultivated tomato sauce affects the clinical progress of systemic nickel allergy syndrome. Observed small differences could not be derived from Ni diet content but from other components that a full food chemical analysis, with detailed methods and results, might reveal. Also missing individual patient data, if disclosed, might be important but, likely, would not be sufficient to support main conclusion, as it is still necessary to improve experimental design to allow to distinguish the specific effect of low Ni diet from that of the tomato preparations and to increase patients number, as it is quite limited.

Author Response

July, 25th 2022

To Editor and Reviewers
Journal of Clinical Medicine MDPI

We would like to greatly thank the Editor and Reviewers who encouraged a further revision of the manuscript.

Please find the enclosed the Revision vers. 2 of the Original article entitled “DATTERINO Trial: a double blind, randomized, controlled, cross-over, clinical trial on the use of Hydroponic Cultivated Tomato Sauce in Systemic Nickel Allergy Syndrome” by Angela Rizzi, Raffaella Chini, Serena Porcari, Carlo Romano Settanni, Eleonora Persichetti, Vincenzina Mora, Caterina Fanali, Alessia Leonetti, Giuseppe Parrinello, Franziska Michaela Lohmeyer, Riccardo Inchingolo, Maria Cristina Mele, Antonio Gasbarrini and Eleonora Nucera.

[Journal of Clinical Medicine] Manuscript ID: jcm-1784715- Major Revisions

Author's Reply to the Review Report (Reviewer 4)
Comments and Suggestions for Authors

While the addition of data about Ni content is important, this does not change the main point impeding the publication of this paper: results are far to be sufficient to conclude that the use of hydroponic cultivated tomato sauce affects the clinical progress of systemic nickel allergy syndrome. Observed small differences could not be derived from Ni diet content but from other components that a full food chemical analysis, with detailed methods and results, might reveal. Also missing individual patient data, if disclosed, might be important but, likely, would not be sufficient to support main conclusion, as it is still necessary to improve experimental design to allow to distinguish the specific effect of low Ni diet from that of the tomato preparations and to increase patients number, as it is quite limited.
We thank the Reviewer for the comments. We agree with the Reviewer on the need to conduct further studies including the determination of metal concentrations in food and biological samples. These research goals now rest on a favourable effect of hydroponic tomato puree on symptom control of SNAS patients.
Individual patient data about treatment adherence were reported in the Figure 4.
Regarding the study design, we didn’t design a comparison among the two tomato sauces and low-Ni diet in allocation ratio 1:1:1, but we compared the two tomato sauces, with low-Ni diet as underlying nutritional treatment of the study in order to avoid patients taking the metal from other food sources. Furthermore, low-Ni diet is a consolidated dietary treatment for SNAS patients.
Finally, the small sample size is well justified because the calculation was based on earlier bibliography, as reported by another Reviewer. We agree with the Reviewer on the need for further studies that support the generalizability of our results.

With the best regards,

Angela Rizzi, Raffaella Chini, Serena Porcari, Carlo Romano Settanni, Eleonora Persichetti, Vincenzina Mora, Caterina Fanali, Alessia Leonetti, Giuseppe Parrinello, Franziska Michaela Lohmeyer, Riccardo Inchingolo, Maria Cristina Mele, Antonio Gasbarrini and Eleonora Nucera

Corresponding Author:
Riccardo Inchingolo, MD, PhD
UOC Pneumologia, Fondazione Policlinico Universitario A. Gemelli IRCCS. Largo A. Gemelli, 8 – 00168 – Rome, Italy.
riccardo.inchingolo@policlinicogemelli.it

Corresponding Author will receive all editorial communications

The authors declare that the manuscript, or specified parts of it, have not been and will not be submitted elsewhere for publication.